# OpenReview forum: "AutoRedTeamer: An Autonomous Red Teaming Agent Against Language Models"
_ICLR.cc/2025/Conference — Submitted to ICLR 2025_

### Official Review · Reviewer_Kor2 · 2024-10-29

**Soundness:** 2
**Presentation:** 3
**Contribution:** 2
**Rating:** 5
**Confidence:** 3

**Summary:**

The paper introduces an automated framework called AutoRedTeamer for red teaming large language models. Unlike existing methods that rely on human intervention, AutoRedTeamer uses an LLM-based agent architecture with five specialized modules and a memory-based attack selection mechanism. This allows for end-to-end automation of seed prompt generation, attack selection, execution, and evaluation. The results show that AutoRedTeamer achieves a 20% higher attack success rate on the HarmBench dataset while reducing computational costs by 46%, and it is able to break defenses and generate test cases comparable to human benchmarks.

**Strengths:**

1. This paper focuses on an important research topic and provides valuable contributions to the community.

2. The paper is well-written and easy to follow overall.

3. The evaluation results demonstrate the potential of the proposed method for effective and efficient automatic red teaming.

**Weaknesses:**

1. The technical contribution, as currently presented, is not significant.

2. The evaluation lacks a detailed description.

**Questions:**

1. The core of the proposed AutoRedTeamer framework is an agent-based ensemble system that dynamically and adaptively selects and refines attack strategies while generating jailbreaking test cases. However, the current text does not elaborate on the technical details of each module in the framework. For instance, when introducing the seed prompt generation module, the authors state that "Each scenario is comprehensively defined, including a unique identifier, detailed description, expected outcome upon target AI failure, and the specific input for the target." However, it is unclear what specific prompts are used to ensure the correctness of the generated seed prompts. The same lack of detail applies to other modules, such as risk analysis and memory-based attack selection. Without these details, it is difficult to quantify the technical contributions of the proposed framework.

2. When evaluating the efficiency of the proposed method, the authors report the total number of queries AutoRedTeamer needs to successfully generate the test cases. However, AutoRedTeamer itself is an LLM-based agent, where each module is an LLM. This means that the overhead cannot simply be measured by the number of queries made to the target model; the time taken by the agent LLM (Mistral-8x22B-Instruct-v0.1) to generate intermediate results should also be considered. Therefore, I suggest that the authors report this overhead (in terms of exact time) and compare it with baseline methods.

3. Several recent LLM jailbreaking techniques shall also be considered as baselines, such as [1]

[1] Zeng, Yi, et al. "How johnny can persuade llms to jailbreak them: Rethinking persuasion to challenge ai safety by humanizing llms." arXiv preprint arXiv:2401.06373 (2024).

---

> ### Author Response · Authors · 2024-11-22
> **response to reviewer**
>
> We thank the reviewer for their thoughtful feedback and for acknowledging the importance of our research topic, the clarity of our writing, and the potential demonstrated by our evaluation results. We address the concerns below.
>
> *Technical contributions unclear*
>
> We appreciate the reviewer's interest in the technical details of each module in our framework. Due to space constraints, we provided the full pseudocode (Appendix C.3) and detailed prompts for each module (Appendix F) in the supplementary material. However, we agree that including more technical details in the main text would enhance the clarity and impact of our work.
>
> * For the Risk Analyzer, we now include more information about how it processes user input to define the scope of red teaming. The Risk Analyzer uses a specialized prompt that instructs the LLM to thoroughly analyze the user's input, breaking it down into core components and identifying potential risks and vulnerabilities. It generates a comprehensive scope of highly specific test cases, focusing on edge cases and scenarios that are most likely to expose vulnerabilities.
> * In the Seed Prompt Generator, we have added details about how it generates diverse test cases covering various aspects such as demographic targets and different scenarios where the risks could manifest. The Seed Prompt Generator uses prompts that instruct the LLM to create test cases that are unique, highly specific, and directly related to the subject matter. It ensures diversity by including different settings, approaches, and targets, and by generating test cases that cover various demographic groups and situations.
> * Regarding the Memory-Based Attack Selection, we have introduced a new subsection in the manuscript to describe the memory component in detail. Our memory system consists of multiple stores: a long-term memory that keeps track of previous attack strategies, their success rates, and their query costs, a long-term memory for each prompt and its success and selected attack, and a short-term memory that records the current test case trajectory.
>
> *Time cost should also be considered*
>
> We agree that evaluating the total computational overhead, including the time taken by the agent LLM modules, is important for a fair assessment of efficiency.
>
> For evaluation on the HarmBench dataset using Llama-3.1-70B as the target model, the results are as follows:
>
> Method | Time Cost | Cost per prompt
> --- | --- | ---
> AutoRedTeamer (Ours) | 4 hours, 25 minutes | 1.1 min
> PAIR | 1 hour, 36 minutes | 0.4 min
> TAP | 6 hours, 14 minutes | 1.6 min
> FewShot | 56 Minutes | 0.23 min
>
> While AutoRedTeamer has a higher per-prompt time compared to PAIR, it achieves higher attack success rates and demonstrates a favorable trade-off between effectiveness and query cost. It is also faster than TAP. Additionally, the per-prompt time remains practical for large-scale evaluations.
>
> *Recent jailbreaking techniques should be considered as baselines*
>
> We appreciate the suggestion to include recent jailbreaking techniques. We update our results below on gpt-4o.
>
> Method | ASR (gpt-4o)
> --- | ---
> Ours | 69%
> PAIR | 53%
> TAP | 66%
> ArtPrompt | 39%
> FewShot | 3%
> Pliny | 37%
> PAP | 18%
>
> Our method outperforms PAP significantly. As a whole, our design is flexible and can easily incorporate new attacks in the toolbox, which we have demonstrated in a separate ablation below where we add the attack on Llama-2-7B from [1] to the toolbox. We find that our framework can also match the 100% success rate by initializing the memory with previously learned success rates. Without this initialized memory, performance is slightly lower as the agent explores less effective strategies before settling on the adaptive attack, which it learns to always select.
>
> Method | ASR
> --- | ---
> Ours with adaptive attack + initialized memory | 100%
> Ours with adaptive attack | 91%
> Ours | 75%
> Adaptive Attack | 100%
>
> We hope our response addressed the reviewer’s concerns. We are happy to answer further questions.
>
> [1] Andriushchenko, M., Croce, F., & Flammarion, N. (2024). Jailbreaking Leading Safety-Aligned LLMs with Simple Adaptive Attacks. ArXiv, abs/2404.02151.

---

> ### Author Response · Authors · 2024-12-01
> **follow up to reviewer**
>
> Dear Reviewer Kor2,
>
> Thank you for taking the time to review our submission and for providing valuable feedback. Since the discussion period is ending, we wanted to see if our response has adequately addressed the concerns and if there are points we could further clarify.

---

### Official Review · Reviewer_Uu9j · 2024-10-31

**Soundness:** 2
**Presentation:** 2
**Contribution:** 1
**Rating:** 3
**Confidence:** 5

**Summary:**

This paper designs an agent to do the automated red-teaming and shows the improved performance over selected baselines.

**Strengths:**

+ The paper is easy to follow
+ The LLM jailbreak topic is trending

**Weaknesses:**

- Marginal novelty
- The improvement is not significant
- Lack of strong baselines
- Need to conduct experiments on more well-aligned models

First of all, I think the idea of using LLM to generate jailbreak prompts is already well-studied since last year, and I do not see significant novelty in this paper compared with tons of similar papers using LLM to mutate/transform/generate new jailbreak prompts. The paper claims that it is the first paper as fully automated, which I cannot agree. Most published black-box jailbreak work just use a set of collected prompts as seeds or just start with some prompts, and then have LLM automatically do the prompt transformation. For example, [1] just starts with some prompts and have LLM automatically do the jailbreak, with is identical to your setting, and I believe there are too many works liks that without human in the loop. Thus, I do not think it is an accurate claim as the first fully automated jailbreak paper.

Second, I think this paper uses agentic approach and combines existing jailbreak strategies while the improvement is not significant, especially considering that there are many black-box attack papers that have near-perfect attack success rates such as [1,2,3].  I think the paper should compare with some latest and stronger attackers since PAIR is published last year and many stronger jailbreak approaches were found after that.

Although I appreciate that the authors conduct experiments on the 70B model, I believe some more aligned models can be attacked with, such as Llama-2, and Claude-3. It would make the experiments more comprehensive with more models in the evaluation.

Lastly, as an attack paper, I think the ethical concern discussion and response disclosure are missing from the manuscript.

[1] GPT-4 Jailbreaks Itself with Near-Perfect Success Using Self-Explanation

[2] WordGame: Efficient & Effective LLM Jailbreak via Simultaneous Obfuscation in Query and Response

[3] Jailbreaking Leading Safety-Aligned LLMs with Simple Adaptive Attacks

**Questions:**

See the comments above

**Details Of Ethics Concerns:**

As an attack paper, I think the ethical concern discussion and response disclosure are missing from the manuscript.

---

> ### Author Response · Authors · 2024-11-22
> **response to reviewer**
>
> We thank the reviewer for their thoughtful review and for acknowledging that our paper is easy to follow and addresses a trending topic in LLM jailbreaks. We address the concerns below.
>
> *Marginal novelty compared to other black-box methods*
>
> We understand the concern regarding the novelty of our work compared to existing black-box attack methods that automate jailbreak prompt generation. While prior works have utilized LLMs to mutate or transform prompts, our contribution lies in the comprehensive, end-to-end automation of the red teaming process with several novel components, rather than proposing a new jailbreaking technique:
> * Automated Seed Prompt Generation and Replacement: Our framework does not rely on manually collected seed prompts, which can limit the diversity and coverage of potential vulnerabilities. Instead, it automatically generates and refines seed prompts based on specified risk categories, ensuring broader exploration of the threat landscape, including less-studied risk categories that may be missed by manual methods. This is the main reason why we claim this is the first fully automated method, as seed prompt generation is an important aspect of red teaming and missed by prior automated techniques, including [1].
> * Integration of Multiple Attacks and Tools: We support a wide range of attacks and mutators within a unified framework, including prior black-box automated techniques such as PAIR, enabling the discovery of synergistic attack combinations. This contrasts with prior works that typically focus on a single attack method or lack the flexibility to incorporate new attacks easily.
>
> We believe these contributions offer a significant advancement over existing black-box attack methods by providing a more adaptive, scalable, and comprehensive approach to automated red teaming.
>
> *Improvement is not significant*
>
> Our framework is designed to be flexible and can easily incorporate new attacks into its toolbox as they are discovered. For instance, we have added the "Pliny" jailbreak [2], a recent and highly effective attack from August 2024, into our evaluation. Our focus is on creating a comprehensive framework that can adapt to and integrate the latest attacks rather than solely achieving higher attack success rates. To illustrate this, we compare our method to individual attacks within the toolbox and show higher performance on red teaming using only weaker attacks.
>
> To further illustrate this idea, we conduct an ablation below where we add the attack on Llama-2-7B from [3] to the toolbox. We find that our framework can also match the 100% success rate by initializing the memory with previously learned success rates. Without this initialized memory, performance is slightly lower as the agent explores less effective strategies before settling on the adaptive attack, which it learns to always select.
>
> Method | ASR
> --- | ---
> Ours with adaptive attack + initialized memory | 100%
> Ours with adaptive attack | 91%
> Ours | 75%
> Adaptive Attack | 100%
>
> *More models should be evaluated*
>
> We appreciate the suggestion. We evaluate Llama-2-7B and Claude-3.5-Sonnet below on HarmBench. We find that similar to other models, AutoRedTeamer outperforms all baselines, including on the newest Claude model, which is very robust.
>
> Method | ASR (Llama-2) | ASR (Claude)
> --- | --- | ---
> Ours | 75% | 28%
> PAIR | 28% | 4%
> ArtPrompt | 19% | 1%
> FewShot | 4% | 0%
>
> *Ethical concern and disclosure are missing*
>
> We agree that discussing ethical considerations is crucial for an attack paper. In the previous manuscript, we have included a broader impact statement in Section B of the Appendix that addresses ethical concerns. We have separated this into an explicit subsection in the revised uploaded version and added a statement about disclosure.
>
> We hope our response addressed the reviewer’s concerns. We are happy to answer further questions
>
> [1] GPT-4 Jailbreaks Itself with Near-Perfect Success Using Self-Explanation
>
>  [2] Pliny the Prompter, "L1B3RT45: Jailbreaks for All Flagship AI Models," 2024.

---

> ### Author Response · Authors · 2024-12-01
> **follow up to reviewer**
>
> Dear Reviewer Uu9j,
>
> Thank you for taking the time to review our submission and for providing valuable feedback. Since the discussion period is ending, we wanted to see if our response has adequately addressed the concerns and if there are points we could further clarify.

---

### Official Review · Reviewer_Uqx7 · 2024-10-31

**Soundness:** 3
**Presentation:** 2
**Contribution:** 2
**Rating:** 3
**Confidence:** 4

**Summary:**

This paper proposes a framework to automatically generate jailbreaks/harmful prompts, evaluate their effectiveness, and iteratively refine the prompts to improve their performance. This is done via a series of LLMs given specific tasks in the overall pipeline. The method does have good performance, generating jailbreaks in fewer queries compared to other methods.

**Strengths:**

- The method is empirically effective - e.g. in Table 1 the ASR is higher (or at least comparable) to the baselines with fewer queries required. Furthermore, from Fig. 8 the fact that many attacks are composed of multiple simple/cheap attack styles does indicate that the method is creating more complex attacks as it iterates over prompts.

- The attack is also robust against defences: for example, the perturbations in SmoothLLM don't significantly degrade attack performance.

-  The method allows fully end-to-end red-teaming with only an initial domain specified (e.g. “cybersecurity”). There is scope here to enhance AutoRedTeamer with attacks tailored to particular use cases that a user is concerned for.

**Weaknesses:**

- Generally, I found the paper to lack detail and technical clarity and the overall paper flow would need improving. Some useful information is for example in the appendix (e.g. pesudocode), which given the low level of technical description in the main body of text could be given a more visible position. Some concrete examples which can use more technical description: in Section 3.2 Risk Analyser, the prompting strategy is not described; alternatively how the memory store works can use more  detail in the paper given its importance to the results and paper claims. Overall, just based on the content of the paper it would be highly challenging to reproduce the results diminishing the contribution of the work.

- The authors raise RedAgent and ALI-agent as similar work using agents to automatically red-team, and additionally state that their use of a memory store is novel. However, those two papers are similar to this one in proposing agentic frameworks and additionally utilise memory. The authors argue that their work is differentiated by the lack of a need for predefined test scenarios and support for a more diverse range of attacks. However, this may not suffice as a compelling degree of novelty - the range of attacks from the other works can be easily extended. Similarly, attacks such as GCG are not compared to because they have fixed test cases: however, in that case, would GCG combined with the seed prompt generation described in the paper be superior to the proposed attack piepline? I feel the authors don’t fully justify why it is so difficult to generate seed prompts and combine it with other attack pipelines.

- Rainbow Teaming is mentioned as the closest work, yet is not evaluated/compared to. Likewise Wildteaming [1] can be considered similar work and could use comparison or at least discussion.

[1] Jiang, Liwei, et al. "WildTeaming at Scale: From In-the-Wild Jailbreaks to (Adversarially) Safer Language Models." arXiv preprint arXiv:2406.18510 (2024).

**Questions:**

- From the transition matrix it seems like many of the attack methods are rarely used. How do you ensure that the strategy designer LLM strikes a balance between exploring different attacks and exploiting successful attacks? Is an LLM the best choice for this as opposed to say RL or similar techniques?

---

> ### Author Response · Authors · 2024-11-22
> **response to reviewer**
>
> We thank the reviewer for the detailed and insightful review. We appreciate the reviewer agreed the method is empirically effective and supported the flexibility of requiring only initial domains.
>
> *Generally, I found the paper to lack detail and technical clarity and the overall paper flow would need improving*
>
> We appreciate the reviewer's concern about insufficient technical descriptions. The paper currently contains key implementation details in the Appendix, including pseudocode (Appendix C.3) and detailed prompts for each module (Appendix F). However, we agree these critical components deserve more prominence in the main paper, so we have moved the pseudocode and added more details to the main body.
>
> Regarding the memory component, we agree the paper would benefit from a more detailed description. We have added a subsection detailing the memory in the methodology. In general, the memory store contains three separate databases: a long-term memory containing previous test cases and selected attacks, a long-term memory containing the query cost and running success rate of each attack, and a short-term memory containing the current trajectory.
>
> *The authors raise RedAgent and ALI-agent as similar work using agents to automatically red-team, and additionally state that their use of a memory store is novel. However, those two papers are similar to this one in proposing agentic frameworks and additionally utilise memory.*
>
> We appreciate the reviewer's thoughtful points about novelty compared to RedAgent and ALI-agent. While these works also leverage agent frameworks, our work makes several key technical advances that enable more effective and comprehensive red teaming of language models.
>
> Indeed, RedAgent and ALI-agent both use agentic frameworks with memory. However, our design is distinguished by the following:
> - The memory designs explored in prior work are simple due to the lack of tools and attacks in these methods, only storing previous test cases and results. In contrast, AutoRedTeamer is differentiated by the comprehensive attack toolbox and attack memory, where the number of attempts, query cost, and running success rate of each explored attack are stored. This allows for continual learning, where attack strategies can be refined over time.
> - In RedAgent, the memory contains previously successful prompts and strategies, and the entire memory store is provided in-context. However, this approach is inefficient and not scalable, as there is a limit to the number of examples that can fit in the prompt. In contrast, AutoRedTeamer uses an embedding-based lookup to identify the most similar prompts, reducing token usage. While ALI-Agent also has a similar embedding-based lookup, its memory is only used for generating new test scenarios, and not for selecting attack strategies.
>
> *the range of attacks from the other works can be easily extended*
>
> We respectfully disagree that it is straightforward to extend the range of attacks in other works. Our work makes several key technical advances that enable more effective integration of diverse attacks:
> - Memory Architecture: Prior works lack a memory component designed for comprehensive attack integration, which we found non-trivial to design. A naive approach of adding attacks would face significant limitations, as the planner would lack contextual information about attack effectiveness, relying solely on textual descriptions that are prone to bias. Our ablation studies in Sec. D demonstrate that our attack memory integration is crucial, improving attack success rates by 26% compared to versions that only use the prior knowledge of the LLM to select attacks.
> - Architectural Limitations: While prior works could theoretically add more attacks, they would require significant architectural changes. RedAgent lacks the necessary modularity and only generates attacks via LLM prompting. They do not demonstrate the ability to incorporate externally defined attacks like those in our toolbox. ALI-agent only uses one attack and would require substantial modifications to support multiple attack types.
>
> Our contribution lies not just in supporting multiple attacks, but in designing a comprehensive framework that can effectively orchestrate them through memory-guided selection and combination. This makes our work fundamentally different from simply adding more attacks to existing methods.

---

> > ### Author Response · Authors · 2024-11-22
> > **response to reviewer (cont)**
> >
> > *Similarly, attacks such as GCG are not compared to because they have fixed test cases…I feel the authors don’t fully justify why it is so difficult to generate seed prompts and combine it with other attack pipelines.*
> >
> > We omit GCG as a baseline as it is a white-box attack, not because it uses fixed test cases. We study a black-box threat model in our paper.
> >
> > While combining seed prompt generation with baseline attacks is technically possible, this misses the key benefits of our comprehensive framework. Our automatic seed prompt generation process removes human bias in test case creation, which is particularly crucial for identifying niche risk categories that might be overlooked in manual curation. We validate this on AIR categories, demonstrating that our automatically generated prompts achieve comparable diversity to human-created benchmarks while covering a broader range of potential risks. This automated approach is increasingly important as LLM-based systems scale, where human review of test cases becomes impractical.
> >
> > Our framework's value comes from the synergistic interaction between modules, particularly in seed prompt generation and attack selection. Rather than using a fixed set of test cases, seed prompts are continuously regenerated and refined if previous ones are unsuccessful, leading to a ~0.30 improvement in success rate compared to the one-step generation, shown in Figure 5. This iterative refinement process leverages our memory architecture to guide the generation of new test cases, creating a more adaptive and comprehensive evaluation framework.
> >
> > *Rainbow teaming and wild teaming is not compared to*
> >
> > We appreciate the reviewer’s suggestion. We compare rainbow teaming below on a subset of 10 level-3 AIR categories on Llama-3.1-70B for 20 iterations for both methods. For a fair comparison, we use the same risk categories and attack styles/mutations.
> >
> > Method | ASR
> > --- | ---
> > Ours | 40%
> > Rainbow Teaming | 18%
> >
> > Our agent-design significantly outperforms the QD search used in rainbow-teaming. This is primarily due to the ability to combine attack vectors; rainbow-teaming is restricted to a grid of one attack per cell.
> >
> > Regarding comparison to WildTeaming, our approach differs fundamentally in both scope and implementation. While WildTeaming focuses specifically on discovering new attack patterns from in-the-wild user interactions, AutoRedTeamer is a comprehensive framework for automated red teaming that integrates diverse existing jailbreak methods and attack vectors into a unified pipeline. Our toolbox architecture allows us to incorporate any effective attack method, including those from WildTeaming.
> >
> > *How do you ensure that the strategy designer LLM strikes a balance between exploring different attacks and exploiting successful attacks?*
> >
> > Ultimately, the goal of the framework is to achieve a high attack success rate at a cheaper cost. To balance exploitation and exploration, our strategy was to emphasize this goal in the instruction of the strategy designer and have it interpret it how it saw fit. This led to exploration where all attacks are tried at least several times (except PAIR, which is generally avoided due to cost) until the agent knows which individual attacks are good and should be selected for future strategies and combinations. When emphasizing exploration in the instruction, this reduced performance.
> >
> > *Is an LLM the best choice for this as opposed to say RL or similar techniques?*
> >
> > We focus on an LLM for several reasons.
> > * Interpretability. Using an LLM offers interpretability in decision-making, mirroring how humans conduct red teaming. This is especially important for safety-critical settings.
> > * Adaptability. While it is possible to train an RL policy to select tools/attacks, the policy will need to be retrained when there is a new attack added to the toolbox.
> > * Practicality. In general, it is easier to use an LLM than to train an RL policy, which requires extensive hyperparameter selection and tuning. An RL policy may also overfit on particular domains in training, and cannot generate seed prompts.
> >
> > We hope this answered the reviewers concerns. We are happy to address further questions.

---

> ### Author Response · Authors · 2024-12-01
> **follow up to reviewer**
>
> Dear Reviewer Uqx7,
>
> Thank you for taking the time to review our submission and for providing valuable feedback. Since the discussion period is ending, we wanted to see if our response has adequately addressed the concerns and if there are points we could further clarify.

---

### Official Review · Reviewer_o3hZ · 2024-11-04

**Soundness:** 2
**Presentation:** 3
**Contribution:** 2
**Rating:** 5
**Confidence:** 4

**Summary:**

This paper proposes an agent-based framework for automatically generating attacks on LLMs, featuring an attack memory bank designed to enhance attack efficiency. The framework demonstrates strong attack performance on HarmBench.

**Strengths:**

1. The design of the risk analyzer within the agent, and the storage format of each entry in the attack memory bank for retrieval purposes are quite interesting.
2. The paper is well-written, with a clear structure. The experiments are extensive.

**Weaknesses:**

1. This paper emphasizes an end-to-end automated red teaming framework; however, the experimental dataset is somewhat limited, lacking exploration of threats in real-world scenarios. It would be more meaningful if the authors could discuss threats in practical contexts, such as embodied scenarios, which would also strengthen their claims about flexibility, adaptability, and scalability.
2. The claims regarding efficiency need further substantiation, as the current experiments do not provide sufficient support.

**Questions:**

1. Why is only HarmBench used in the evaluation? This alone is insufficient to demonstrate the proposed attack framework's flexibility, adaptability, and scalability. Please consider exploring some real-world security threat scenarios, which are more complex and meaningful than the questions in HarmBench.
2. While total queries and total tokens provide insights into efficiency, I am more interested in how long it takes for the end-to-end attack framework to generate an effective attack prompt. Since the end-to-end agent consists of multiple modules, there may be delays in communication between these components, which could negatively impact efficiency. This complexity may be a disadvantage compared to previous methods that do not involve such intricate module interactions.

---

> ### Author Response · Authors · 2024-11-22
> **Response to reviewer**
>
> We thank the reviewer for the insightful review. We appreciate that the reviewer found the design of the agent interesting and paper well-written. We address the concerns below.
>
> *the experimental dataset is somewhat limited, lacking exploration of threats in real-world scenarios…*
>
> Our evaluation is actually quite comprehensive, using multiple datasets that cover a broad range of real-world scenarios:
> - Standard Evaluation: We evaluate our method on HarmBench, which is a standard benchmark for jailbreaking methods. This allows direct comparison with prior work.
> - Diverse Risk Categories: We also evaluate on the AIR taxonomy and AIR-Bench, which contain a broader range of risk categories missing from HarmBench, such as Autonomous Operation of Systems and Political Persuasion. This demonstrates the comprehensive design of our framework.
> - Our coverage of risk categories is derived directly from 2024 government regulations like the EU AI Act, ensuring our evaluation addresses current real-world safety concerns. This comprehensive evaluation distinguishes our work from prior papers [1, 2] that evaluate only a single dataset.
>
> Regarding embodied scenarios: While these are emerging and interesting, we note that LLM agents are fundamentally extensions of LLMs. Establishing robustness in text-based scenarios is a necessary foundation before moving to more complex embodied contexts. We have expanded the discussion in the revision and identified embodied scenarios as a promising direction for future work.
>
> *I am more interested in how long it takes for the end-to-end attack framework to generate an effective attack prompt.*
>
> Thanks for the suggestion. We provide the results on the time cost of our framework below on Llama-3.1-70B.
>
> Method | Time Cost | Cost per prompt
> --- | --- | ---
> AutoRedTeamer (Ours) | 4 hours, 25 minutes | 1.1 min
> PAIR | 1 hour, 36 minutes | 0.4 min
> TAP | 6 hours, 14 minutes | 1.6 min
> FewShot | 56 Minutes | 0.23 min
>
> We find that optimization-based methods, similar to query cost, also have a larger time cost. However, the cost of generating a single prompt takes around a minute for all methods, which is fast. AutoRedTeamer takes longer than PAIR despite being more query-efficient, but is much faster than TAP.
>
> We hope this has resolved the reviewer’s concerns. We are happy to address further questions.
>
> [1] Automated Red Teaming with GOAT: The Generative Offensive Agent Tester https://openreview.net/forum?id=Ly0SQh7Urv
>
> [2] Does Refusal Training in LLMs Generalize to the Past Tense? https://openreview.net/forum?id=aJUuere4fM

---

> > ### Comment · Reviewer_o3hZ · 2024-11-27
> >
> > Thank you to the authors for their response. However, after careful consideration, I have concerns regarding the utility and persuasiveness of the issues discussed in the paper. I will maintain my current rating.

---

### Author Response · Authors · 2024-11-22
**General response**

We thank the reviewers for their valuable feedback and comments. We are glad that the reviewers find our paper well-written (Reviewer o3hZ, Uu9j, Kor2), topic important (Reviewer Uu9j, Kor2), framework design interesting (Reviewer 03hZ), and acknowledge the empirical effectiveness of our method (Reviewer Uqx7, Kor2). We especially appreciate the recognition of our framework's potential for effective and efficient automatic red teaming (Reviewer Uqx7). We provide the summary of our revision content below to further strengthen our paper.

1. Our main contribution is providing the first comprehensive, automated red teaming framework. We demonstrate this comprehensiveness in several ways:
   - Flexibility across domains: Unlike prior work that only evaluates on fixed test cases, we show effectiveness on both existing input prompts from benchmarks (HarmBench) and diverse risk categories (AIR taxonomy)
   - Automated seed prompt generation: We demonstrate that our framework can match human-level diversity in test case creation while removing potential human biases
   - Integration of multiple attack vectors: Our framework can incorporate and intelligently combine various attacks, as shown by our successful integration of recent methods like Pliny

2. Following reviewers' suggestions, we have conducted additional experiments to strengthen our claims:
   - Added time efficiency results showing practical runtime despite comprehensive automation
   - Evaluated on more recent models (Claude-3.5-Sonnet, Llama-2-7B)
   - Included comparisons to 2024 attacks such as the highly effective adaptive attack [1], showing our framework can match or exceed their performance when incorporated into our toolbox
   - Added comparison to other end-to-end pipelines such as rainbow teaming [2], demonstrating significant improvements

3. We have added more technical details about our modules' implementation and enhanced our discussion of ethical considerations. We will release our code to aid reproducibility and future research.

We believe these revisions address the main concerns raised while reinforcing our paper's key contribution: a flexible, comprehensive framework that advances the state of automated red teaming.

[1] Andriushchenko, M., Croce, F., & Flammarion, N. (2024). Jailbreaking Leading Safety-Aligned LLMs with Simple Adaptive Attacks. ArXiv, abs/2404.02151.
[2] Samvelyan, M., Raparthy, S.C., Lupu, A., Hambro, E., Markosyan, A.H., Bhatt, M., Mao, Y., Jiang, M., Parker-Holder, J., Foerster, J., Rocktaschel, T., & Raileanu, R. (2024). Rainbow Teaming: Open-Ended Generation of Diverse Adversarial Prompts. ArXiv, abs/2402.16822.

---

### Meta-Review · Area_Chair_T6D3 · 2024-12-15

**Metareview:**

The AC and reviewers acknowledge that the authors have addressed some concerns, such as providing additional technical clarifications to enhance the paper. However, significant concerns remain regarding the utility and persuasiveness of the issues discussed. Additionally, both the AC and reviewers question the novelty of the automated red-teaming approach, particularly in light of the substantial body of similar research in the field. The primary contribution of the proposed work—focused on improving attack capability—appears aligned with the objectives of many existing studies, which raises further concerns about its novelty from the scope perspective.

**Additional Comments On Reviewer Discussion:**

The reviewer raised concerns about the flexibility, adaptability, and scalability of the proposed attack framework, noting that the authors did not explore real-world security threat scenarios. Specifically, HarmBench was deemed unconvincing as a representation of real-world threats. In their response, the authors emphasized the superiority of HarmBench but did not address the need for more realistic threat scenarios. The AC recommends taking additional steps to explore scenarios that are more complex and meaningful than the questions posed in HarmBench, to better demonstrate the framework's applicability in real-world contexts.

---

### Decision · Program_Chairs · 2025-01-22

Reject